# Clinicopathological Factors Related to Recurrence Patterns of Resected Non-Small Cell Lung Cancer

**DOI:** 10.3390/jcm9082473

**Published:** 2020-08-01

**Authors:** Reiko Shimizu, Tomomari Kinoshita, Naomichi Sasaki, Mao Uematsu, Yusuke Sugita, Toshiyuki Shima, Masahiko Harada, Tsunekazu Hishima, Hirotoshi Horio

**Affiliations:** 1Department of Thoracic Surgery, Tokyo Metropolitan Cancer and Infectious Diseases Center Komagome Hospital, Tokyo 113-8677, Japan; reiko.0908.reiko@gmail.com (R.S.); chimiona.1025@gmail.com (N.S.); m.uematsu0718@gmail.com (M.U.); y.sugita07@gmail.com (Y.S.); hitorimatatabi1983@gmail.com (T.S.); msharada@cick.jp (M.H.); hirohori@cick.jp (H.H.); 2Department of Pathology, Tokyo Metropolitan Cancer and Infectious Diseases Center Komagome Hospital, Tokyo 113-8677, Japan; hishima@cick.jp

**Keywords:** clinicopathological factors, metastasis sites, non-small cell lung cancer

## Abstract

Even after complete resection, non-small cell lung cancer (NSCLC) shows preferential recurrence in the mediastinal lymph nodes, lungs, brain, bone, liver, and adrenal gland. However, the relationship between clinicopathological factors and recurrence patterns after resection has not been well-evaluated. Among 688 NSCLC cases with complete resection between 2004 and 2016, 233 cases recurred at our institute. On multivariate analyses, NSCLCs with lymph node metastasis and pulmonary metastasis at surgery commonly recurred in the mediastinal lymph nodes and lungs, respectively. Young age, adenocarcinoma, and vascular invasion were correlated with brain metastasis. Although no variable was associated with bone metastasis, vascular invasion was correlated with postoperative liver and adrenal gland metastasis. Pathologically proven stage II or III NSCLC, adenocarcinoma, and the presence of lymphatic permeation would result in multiple metastases. Vascular invasion, larger invasive size, and advanced stage were independent risk factors of early recurrence. Considering survival, vascular invasion, elderly age, and non-adenocarcinoma were unfavorable prognostic factors after recurrence. Some clinicopathological variables were correlated with organ-specific metastasis and post-recurrence survival. Particularly, vascular invasion was a biomarker of brain, liver, and adrenal gland metastases and a prognostic marker after recurrence among completely resected NSCLC. This information is useful for more frequent patient follow-up and identifying organ-specific distant metastasis.

## 1. Introduction

Owing to the use of high-resolution computed tomography (HRCT), small and resectable non-small cell lung cancer (NSCLC) are increasingly being detected worldwide [1,2]. Nevertheless, NSCLC recurrence, even after complete resection, is still frequently observed in clinical settings [3,4]. Many studies have identified some clinicopathological factors that could predict the postoperative prognosis of patients with NSCLC [5,6,7,8,9]. Thus, identifying predictors of recurrence in patients with completely resected NSCLC would be very helpful in determining whether a shorter period systemic evaluation should be performed during follow-up.

Solid tumors have great variations in the patterns of metastatic organ tropism. In other words, a particular cancer tends to relapse to one particular organ or multiple specific organ sites. In general, NSCLC is likely to metastasize to some organs, including the mediastinal lymph nodes, lungs, brain, bones, liver, and adrenal glands [10,11,12,13,14]. Although many studies evaluated the relationship between clinicopathological factors and postoperative recurrence, the patterns of recurrence after complete resection have not been well described. Therefore, in the current study, we evaluated the relationship between perioperative clinicopathological factors and recurrence site patterns after complete resection and determined the correlation between these factors and survival after recurrence.

## 2. Materials and Methods

### 2.1. Patient Characteristics

Between January 2004 and December 2016, a total of 688 patients with NSCLC underwent complete resection and systematic lymph node dissection at our institute. For diagnostic work-up, all patients underwent chest contrast-enhanced HRCT and positron emission tomography (PET) CT. When nodal metastasis was suspected, additional examinations were performed, such as endobronchial ultrasound-guided transbronchial needle aspiration or biopsy using mediastinoscopy. All therapies, including surgery, adjuvant chemotherapy, and post-recurrent therapies, were discussed and decided at a multi-disciplinary team conference. Among them, 223 patients had a postoperative recurrence of NSCLC. Patients were excluded if they underwent sublobar resection and/or neoadjuvant chemotherapy and had NSCLC with positive intraoperative pleural lavage cytology. The detailed data of these enrolled patients are shown in Table 1.

Data collection and analyses were approved by our institutional review board (Ethical Review Board at Komagome hospital) in January 2020 (approval number: 2482). As the research was a retrospective review of chart data and specimens, and no personally identifiable information was included, the need to obtain written informed consent was waived.

### 2.2. Pathological Evaluation

We evaluated all the pathology slides of resected specimens. After fixing the specimens with 10% formalin and embedding them in paraffin, serial 4 μm sections were stained with hematoxylin/eosin and elastica van Gieson to assess the degree of pleural and vascular invasion. When required, we also stained the sections with D2-40 to evaluate the extent of lymphatic permeation. The cases were reviewed according to the 4th edition of the World Health Organization histological classification and staged according to the 8th edition of the TNM classification of the Union for International Cancer Control.

### 2.3. Follow-Up

All patients were followed-up at our outpatient department quarterly in the first two years after resection and semi-annually thereafter. Contrast-enhanced CT scans of the chest and upper abdomen were routinely obtained during every scheduled outpatient visit for follow-up. CT or magnetic resonance imaging (MRI) of the brain were also routinely performed semi-annually. Such radiological examinations were also performed when neurological symptoms occurred or when clinical suspicions were raised. Once any metastasis was discovered, a routine examination such as PET was performed to determine the presence of other metastatic sites, if any.

Local recurrence was defined as tumor recurrence in contiguous anatomical sites, including the ipsilateral hemithorax and mediastinum after resection. Distant metastasis was defined as tumor recurrence in the contralateral lung or outside the hemithorax and mediastinum after resection. If both local and distant recurrences were noted within three months at the time of initial recurrence, these cases were classified into the distant metastasis group or the multiple metastatic group in number analysis (single vs. multiple). The location of recurrence at the time of final recurrence was defined as all organs where metastatic tumors were identified by the time of death or the last follow-up, regardless of the treatment effect. For patients who did not undergo resection or biopsy to confirm tumor recurrence, judgment was made according to the clinical course, progression, or aggressive clinical behavior.

### 2.4. Statistical Analysis

The length of organ-specific metastasis-free survival (time to recurrence) was defined as the interval between the date of resection and the date of the specific organ sites of metastasis (i.e., mediastinal lymph nodes, lungs, brain, bones, or liver) or the last follow-up. The length of survival after recurrence was defined as the interval between the date when recurrence was identified and the date of death or the last follow-up. Survival curves were drawn using the Kaplan-Meier method. The log-rank test was used to perform comparisons. To investigate the association between clinicopathological factors and specific organ sites of metastasis, clinicopathological factors were analyzed using the chi-squared/Fisher tests and multivariate logistic regression test. For specific organ site metastasis-free survival, univariate and multivariate analyses were performed using the Cox proportional hazards model. All variables with *p* < 0.1 on univariate analysis were entered into the multivariate forward-backward stepwise model. Statistical significance was defined as *p* < 0.05. All analyses were performed using SPSS software (version 25; IBM, Armonk, NY, USA).

## 3. Results

### 3.1. Organ-specific Recurrence

Considering organ-specific recurrence, the results of univariate (chi-squared tests) and multivariate (multiple logistic regression tests) analyses in all cases (*n* = 688) and recurrence cases (*n* = 223) are shown in Table 2, Table 3, Table 4, Table 5, Table 6 and Table 7 and Appendix A. Both at the time of initial and final recurrence, node-positive NSCLCs recurred more often in the mediastinal lymph nodes (HR: 5.46, *p* = 0.001; and HR: 4.59, *p* = 0.001 in all cases, HR: 2.98, *p* = 0.001; and HR: 2.45, *p* = 0.002 in recurrence cases, respectively), and NSCLCs with pulmonary metastasis at surgery relapsed significantly more often in the lungs (HR: 2.11, *p* = 0.042; and HR: 2.01, *p* = 0.044 in all cases, HR: 2.80, *p* = 0.041; and HR: 2.82, *p* = 0.046 in recurrence cases, respectively). On evaluating typical distant organ metastasis, the factors correlated with brain metastasis both at initial and final recurrence included young age (HR: 2.39, *p* = 0.010; and HR: 1.98, *p* = 0.017 in all cases, HR: 2.28, *p* = 0.021; and HR: 2.09, *p* = 0.017 in recurrence cases, respectively), adenocarcinoma (HR: 2.65, *p* = 0.018; and HR: 4.37, *p* = 0.001 in all cases, HR: 2.45, *p* = 0.038; and HR: 4.72, *p* = 0.001 in recurrence cases, respectively), and tumors with vascular invasion (HR: 6.45, *p* = 0.001; and HR: 4.27, *p* = 0.001 in all cases, HR: 3.09, *p* = 0.029; and HR: 3.07, *p* = 0.006 in recurrence cases, respectively). Although no variable was associated with bone metastasis in recurrence cases, vascular invasion in surgical specimens was associated with postoperative liver metastasis both at initial and final recurrence (HR: 9.85, *p* = 0.001; and HR: 7.58, *p* = 0.007 in all cases, HR: 5.85, *p* = 0.001; and HR: 2.59, *p* = 0.055 in recurrence cases, respectively) and adrenal gland metastasis (HR: 8.86, *p* = 0.001; and HR: 7.90, *p* = 0.001 in all cases, HR: 6.86, *p* = 0.001; and HR: 7.95, *p* = 0.001 in recurrence cases, respectively) (Table 2, Table 3, Table 4, Table 5, Table 6 and Table 7 and Appendix A).

### 3.2. Location (Local vs. Distant) and Numbers (Single vs. Multiple) of Recurrence Sites

We focused on the recurrence cases hereafter. The clinicopathological factors and location (local vs. distant) at the time of initial recurrence were not significantly correlated (Table 8). However, multiple metastases at the time of initial recurrence were observed after resection of advanced stage NSCLC (hazard ratio (HR): 2.57, *p* = 0.012). On evaluating all recurrence sites at the time of final observation, adenocarcinoma (HR: 2.06, *p* = 0.016) and the presence of lymphatic permeation (HR: 1.75, *p* = 0.048) were risk factors for multiple recurrence (Table 9).

### 3.3. Time to Recurrence

Among the cases with recurrence, some clinicopathological factors were correlated with early recurrence, as shown in Table 10 and Appendix A and Figure 1. On multivariate analysis, the presence of vascular invasion (HR: 1.56, *p* = 0.008), larger invasive size (HR: 1.37, *p* = 0.028), and advanced stage (HR: 1.43, *p* = 0.027) were independent risk factors of early recurrence among recurrent cases (Table 4 and Appendix A).

### 3.4. Survival after Recurrence

In addition to time to recurrence, vascular invasion was an unfavorable prognostic factor even after recurrence on multivariate analysis (HR: 1.61, *p* = 0.048; Table 4 and Appendix A and Figure 1). Old age and non-adenocarcinoma cases were also poor prognostic factors (HR: 1.99, *p* = 0.001; and HR: 1.95, *p* = 0.001, respectively)

## 4. Discussion

The results of the current study demonstrated a strong correlation between some perioperative clinicopathological factors and recurrence patterns including the location and number of metastases at the time of initial and final recurrence. Although a couple of studies have shown various predictors of brain metastasis after complete resection, few studies have shown that NSCLC preferentially relapses to other organs including mediastinal lymph nodes, liver, and adrenal gland on systemic evaluation of recurrence sites [15,16,17,18]. Hung et al. demonstrated that the pathological subtypes of lung adenocarcinoma are associated with organ-specific metastasis in patients with resected lung adenocarcinoma with distant metastasis [19]. They included all surgery cases regardless of recurrence into survival analyses to calculate the HR of organ-specific recurrence and reported some clinicopathological factors that were independently correlated with organ-specific recurrence. For determining concise and true risk factors of organ-specific recurrence, only cases of recurrence should be also analyzed, as was done in the current study because almost all resected tumors without recurrence during a certain observation period after surgery may not have potential to metastasize. It would be very important and intriguing to analyze those factors among recurrence cases, which definitely have potential for metastasis. Moreover, most studies focused only on the site of initial recurrence; they did not focus on the organs showing late recurrence. In the current study, we distinguished between initial and final recurrence sites. Although there were no apparent differences in the risk factors for the initial and final recurrence sites, to the best of our knowledge, the current study is the first to evaluate the metastatic lesions until the end of the follow-up period.

The mediastinal lymph nodes, lungs, brain, bones, adrenal gland are the most common organ sites of metastasis of resected NSCLC [10,11,12,13,14]. Regarding intrathoracic recurrence, mediastinal lymph node recurrence is associated with node-positive NSCLC resection whereas pulmonary metastasis at the time of surgery was associated with lung recurrence. These two sites, i.e., the mediastinal lymph nodes and lungs, were the most common sites of metastases in the current study. Enhanced CT was helpful for detecting these types of metastases during follow-up. Brain metastasis is the most serious issue in patients because it contributes to the decrease in the quality of life. In fact, some studies have evaluated various predictors for brain metastasis after complete resection [15,16,17,18]. For example, Hubbs et al. showed that young age, larger tumor size, lymphovascular invasion, and hilar lymph node involvement were associated with an increased risk of brain metastasis [15]. Although this result is very similar to those obtained in the current study, these factors were compared among all resected cases (not only recurrence cases) and vascular invasion and/or lymphatic permeation was considered ‘lymphovascular invasion’ [7]. On survival analyses as well as on evaluating the risk factors for organ-specific recurrence, vascular invasion and lymphatic permeation have different significance as risk factors, suggesting that these two pathological factors should be distinguished considering the biology of NSCLC. Among cases of adenocarcinoma, micropapillary-predominant adenocarcinoma was significantly associated with a higher incidence of brain metastasis [19]. In the current study, no such tendency was found, and only four patients had micropapillary-predominant adenocarcinoma. The incidence of adenocarcinoma subtypes may differ among countries. Anyway, this information is important and helpful for identifying patients with resected lung adenocarcinoma who are at a higher risk of developing brain metastases because early diagnosis of brain metastasis is difficult in patients without neurologic symptoms.

While the liver was not the most common organ site of metastasis of lung cancer, liver metastasis was observed during follow-up after surgery. Few studies have evaluated liver metastasis after the resection of NSCLC. Vascular invasion was an independent risk factor for liver metastasis as well as brain and adrenal gland metastasis. Small malignant nodules in the liver are usually difficult to differentiate from benign liver cysts. Hence, if follow-up CT reveals a liver nodule after the resection of NSCLC with vascular invasion, additional examination is required.

Regarding numbers of recurrence sites, even though tumor relapsed postoperatively, single recurrence site was found more frequently among non-adenocarcinomas, NSCLCs without lymphatic permeation, and those with lymph node metastasis at surgery when observed for long period as shown in Table 3. Locoregional treatment such as surgery and radiation for such cases would be effective to control and manage recurred tumor.

Among metastatic cases of NSCLC, vascular invasion, large invasive size, and advanced stage were independent risk factors of early recurrence. Although these factors and others—such as lymphatic permeation and pleural invasion—are known prognostic factors for recurrence-free survival, the risk factors for early recurrence have not been evaluated previously [5,6,7,8,9]. Cases of NSCLC with vascular invasion would require frequent examination including systemic radiological evaluation. Moreover, vascular invasion is significantly associated with shorter survival after recurrence. However, till date, no biomarker has been found for post-recurrence survival. Hence, careful follow-up after recurrence is required in cases with vascular invasion.

The current study has some limitations and biases. Because the study was retrospective and performed at a single institute, patient selection bias and time trend bias were inevitable. Genetic information such as driver mutations and immunogenic status including programmed cell death 1/programmed cell death 1 ligand 1 status were not evaluated since not all cases had this information. In addition, the study might have had diagnostic bias, as conventional imaging including full-body CT could not always identify all metastatic lesions. There was also bias in distinguishing second primary lung cancer from recurrent NSCLC. Last, although cases with postoperative radiation for prevention of local recurrence were not included, clinical effect of postoperative systemic chemotherapy was not considered because systemic treatment would not change preferential patterns of recurrence.

## 5. Conclusions

In conclusion, the results of the current study demonstrate that some clinicopathological factors were associated with organ-specific metastasis, number of metastasis sites, and post-recurrence survival. This information is important for determining which patients need follow-up with shorter period systemic evaluation and further studies regarding the molecular mechanisms that lead to organ-specific metastasis in NSCLC.

## Figures and Tables

**Figure 1 jcm-09-02473-f001:**
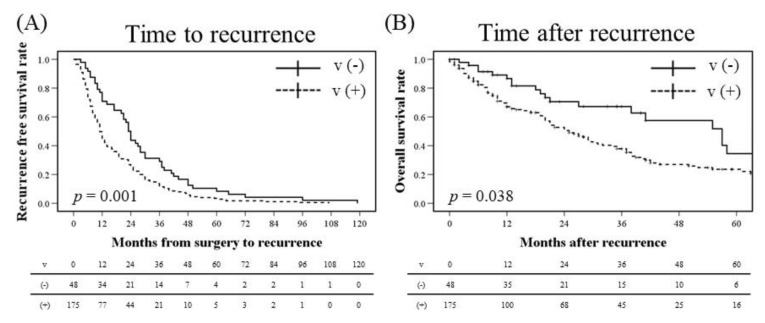
Survival curves of time to recurrence (**A**) and time after recurrence (**B**), according to vascular invasion. V, vascular invasion.

**Table 1 jcm-09-02473-t001:** Clinicopathologic Characteristics of Patients (All cases and recurrence cases).

Factors	All Cases *n* = 688 (%)	Recurrence Cases *n* = 223 (%)
sex male/female	408/280 (59/41)	145/78 (65/35)
age median (range)	71 (19–91)	69 (19–87)
observation period mean months (range)	57.3 (4–194)	58.5 (4−194)
smoking ever/never	482/206 (70/30)	164/59 (74/36)
surgery lobectomy/bilobectomy/pneumonectomy	653/30/5 (95/4/1)	208/12/3 (93/5/2)
histological type AD/SQ/ADSQ/LA/LCNEC/others	470/163/14/11/13/17 (68/24/2/2/2/2)	154/45/6/3/5/10 (69/20/3/1/2/5))
invasive size <3cm/3–5cm/5–7cm/7cm>	409/202/55/22 (59/30/8/3)	85/96/28/14 (38/43/13/6)
lymphatic permeation present/absent	307/381 (45/55)	120/103 (54/46)
vascular invasion present/absent	386/302 (56/44)	175/48 (78/22)
pleural invasion present/absent	143/545 (21/79)	130/93 (58/42)
pulmonary metastasis present/absent	41/647 (6/94)	18/205 (8/92)
lymph node metastasis N0/N1/N2	496/90/102 (72/13/15)	107/44/72 (48/20/32)
adjuvant therapy performed/undone	170/518 (25/75)	88/135 (39/61)
pathological stage IA/IB/IIA/IIB/IIIA/IIIB	228/151/44/128/99/38 (33/22/6/19/14/6)	22/41/13/59/58/30 (10/18/6/26/26/14)

AD, adenocarcinoma; SQ, squamous cell carcinoma; ADSQ, adenosquamous carcinoma; LA, large cell carcinoma; LCNEC, large cell neuroendocrine carcinoma.

**Table 2 jcm-09-02473-t002:** Relationship between clinicopathologic factors and organ-specific recurrence (mediastinal lymph node) on multivariate logistic regression test.

Mediastinal Lymph Node Metastasis									
Factors		All Cases				Recurrence Cases			
		Initial Recurrence		Final Recurrence		Initial Recurrence		Final Recurrence	
		HR (95% CI)	*p*	HR (95% CI)	*p*	HR (95% CI)	*p*	HR (95% CI)	*p*
Sex	male	1.92 (1.03–3.56)	0.004	-		2.11 (1.11–4.05)	0.025	-	
	female	1				1			
Pleural invasion	present	3.01 (1.72–5.26)	0.001	3.48 (2.05–5.92)	0.001	-		-	
	absent	1		1					
Invasive size	>3 cm	1.81 (1.02–3.21)	0.043	1.94 (1.14–3.30)	0.014	-		-	
	≤3 cm	1		1					
Lymph node metastasis	present	5.46 (3.09–9.62)	0.001	4.59 (2.70–7.81)	0.001	2.98 (1.62–5.50)	0.001	2.45 (1.38–4.35)	0.002
	absent	1		1		1		1	

**Table 3 jcm-09-02473-t003:** Relationship between clinicopathologic factors and organ-specific recurrence (lung) on multivariate logistic regression test.

Lung Metastasis									
Factors		All Cases				Recurrence Cases			
		Initial Recurrence		Final Recurrence		Initial Recurrence		Final Recurrence	
		HR (95% CI)	*p*	HR (95% CI)	*p*	HR (95% CI)	*p*	HR (95% CI)	*p*
Vascular invasion	present	-		-		0.49 (0.26–0.96)	0.037	-	
	absent					1			
Pleural invasion	present	6.62 (3.91–11.2)	0.001	7.46 (4.61–12.1)	0.001	-		-	
	absent	1		1					
Pulmonary metastasis	present	2.11 (1.01–4.88)	0.042	2.01 (1.01–4.46)	0.044	2.80 (1.04–2.80)	0.041	2.82 (1.02–7.81)	0.046
	absent	1		1		1		1	
Invasive size	>3 cm	2.33 (1.35–4.03)	0.002	2.03 (1.25–3.30)	0.004	-		-	
	≤3 cm	1		1					

**Table 4 jcm-09-02473-t004:** Relationship between clinicopathologic factors and organ-specific recurrence (brain) on multivariate logistic regression test.

Brain Metastasis									
Factors		All Cases				Recurrence Cases			
		Initial Recurrence		Final Recurrence		Initial Recurrence		Final Recurrence	
		HR (95% CI)	*p*	HR (95% CI)	*p*	HR (95% CI)	*p*	HR (95% CI)	*p*
Age	≥70	0.42 (0.22–0.81)	0.010	0.51 (0.29–0.87)	0.017	0.44 (0.22–0.88)	0.021	0.48 (0.26–0.88)	0.017
	<70	1		1		1		1	
Pathology	AD	2.65 (1.18–5.96)	0.018	4.37 (2.10–9.06)	0.001	2.45 (1.05–5.68)	0.038	4.77 (2.20–10.1)	0.001
	others	1		1		1		1	
Vascular invasion	present	6.45 (2.42–17.2)	0.001	4.27 (2.02–9.01)	0.001	3.09 (1.12–8.48)	0.029	3.07 (1.38–6.80)	0.006
	absent	1		1		1		1	
Pleural invasion	present	2.68 (1.40–5.13)	0.003	4.41 (2.53–7.69)	0.001	-		-	
	absent	1		1					
Pathological stage	II–III	-		2.11 (1.13–3.96)	0.010	-		-	
	I			1					

**Table 5 jcm-09-02473-t005:** Relationship between clinicopathologic factors and organ-specific recurrence (bone) on multivariate logistic regression test.

Bone Metastasis									
Factors		All Cases				Recurrence Cases			
		Initial Recurrence		Final Recurrence		Initial Recurrence		Final Recurrence	
		HR (95% CI)	*p*	HR (95% CI)	*p*	HR (95% CI)	*p*	HR (95% CI)	*p*
Vascular invasion	present	2.95 (1.08–8.07)	0.037	2.45 (1.21–4.95)	0.013	-		-	
	absent	1		1					
Pleural invasion	present	5.32 (2.53–11.2)	0.001	4.00 (2.24–7.14)	0.001	-		-	
	absent	1		1					

**Table 6 jcm-09-02473-t006:** Relationship between clinicopathologic factors and organ-specific recurrence (liver) on multivariate logistic regression test.

Liver Metastasis									
Factors		All Cases				Recurrence Cases			
		Initial Recurrence		Final Recurrence		Initial Recurrence		Final Recurrence	
		HR (95% CI)	*p*	HR (95% CI)	*p*	HR (95% CI)	*p*	HR (95% CI)	*p*
Vascular invasion	present	9.85 (3.90–58.6)	0.001	7.58 (1.74–33.3)	0.007	5.85 (1.86–12.5)	0.001	2.59 (0.96–7.45)	0.055
	absent	1		1		1		1	
Pleural invasion	present	3.257 (1.33–7.94)	0.010	2.70 (1.20–6.06)	0.016	-		-	
	absent	1		1					

**Table 7 jcm-09-02473-t007:** Relationship between clinicopathologic factors and organ-specific recurrence (adrenal gland) on multivariate logistic regression test.

Adrenal Gland Metastasis									
Factors		All Cases				Recurrence Cases			
		Initial Recurrence		Final Recurrence		Initial Recurrence		Final Recurrence	
		HR (95% CI)	*p*	HR (95% CI)	*p*	HR (95% CI)	*p*	HR (95% CI)	*p*
Vascular invasion	present	8.86 (3.14–53.5)	0.001	7.99 (3.42–40.5)	0.001	6.86 (2.15–13.5)	0.001	7.95 (3.23–20.5)	0.001
	absent	1		1		1		1	
Pleural invasion	present	3.32 (1.03–10.6)	0.044	3.45 (1.28–9.35)	0.014	-		-	
	absent	1		1					

AD, adenocarcinoma; HR, Hazard ratio; 95% CI, 95% confidence interval.

**Table 8 jcm-09-02473-t008:** The Relationship between clinicopathological factors and recurrence patterns (locations and numbers) at initial recurrence and final recurrence on chi-squared/Fisher test.

Factors		Initial Recurrence						Final Recurrence		
		Location			Number			Number		
		Local (%)	Distant (%)	*p*	Single (%)	Multiple (%)	*p*	Single (%)	Multiple (%)	*p*
Sex	male	64 (44)	81 (56)	0.079	97 (67)	48 (33)	0.580	59 (41)	86 (59)	0.891
	female	25 (32)	53 (68)		55 (71)	23 (29)		31 (40)	47 (60)	
Age	≥70	48 (44)	62 (56)	0.262	72 (65)	38 (35)	0.392	51 (46)	59 (54)	0.071
	<70	41 (36)	72 (64)		80 (71)	33 (29)		39 (35)	74 (65)	
Smoking	ever	63 (38)	101 (62)	0.447	112 (68)	52 (32)	0.944	68 (41)	96 (59)	0.575
	never	26 (44)	33 (56)		40 (68)	19 (32)		22 (37)	37 (63)	
Pathology	AD	56 (36)	98 (64)	0.106	103 (67)	51 (33)	0.540	52 (34)	102 (66)	0.003
	others	33 (48)	36 (52)		49 (71)	20 (29)		38 (55)	31 (45)	
Lymphatic permeation	present	46 (38)	74 (62)	0.604	78 (65)	42 (35)	0.274	40 (33)	80 (67)	0.021
	absent	43 (42)	60 (58)		74 (72)	29 (28)		50 (49)	53 (51)	
Vascular invasion	present	65 (37)	110 (63)	0.107	115 (66)	60 (34)	0.134	68 (39)	107 (61)	0.383
	absent	24 (50)	24 (50)		37 (77)	11 (23)		22 (46)	26 (54)	
Pleural invasion	present	53 (41)	77 (59)	0.757	89 (68)	41 (32)	0.909	55 (42)	75 (58)	0.483
	absent	36 (39)	57 (61)		63 (68)	30 (32)		35 (38)	58 (62)	
Pulmonary metastasis	present	6 (33)	12 (67)	0.552	8 (44)	10 (56)	0.024	5 (28)	13 (72)	0.256
	absent	83 (40)	122 (60)		144 (70)	61 (30)		85 (41)	120 (59)	
Invasive size	>3cm	54 (39)	84 (61)	0.762	91 (66)	47 (34)	0.365	59 (43)	79 (57)	0.353
	≤3cm	35 (41)	50 (59)		61 (72)	24 (28)		31 (50)	31 (50)	
Lymph node metastasis	present	46 (40)	70 (60)	0.935	74 (64)	42 (36)	0.145	51 (48)	56 (52)	0.033
	absent	43 (40)	64 (60)		78 (73)	29 (27)		39 (34)	77 (66)	
Adjuvant therapy	performed	53 (60)	35 (40)	0.973	60 (68)	28 (32)	0.996	34 (39)	54 (61)	0.672
	undone	81 (60)	54 (40)		92 (68)	43 (32)		56 (41)	79 (59)	
Pathological stage	II–III	65 (41)	95 (59)	0.728	100 (63)	60 (38)	0.004	59 (37)	101 (63)	0.091
	I	24 (38)	39 (62)		52 (83)	11 (17)		31 (49)	32 (51)	

**Table 9 jcm-09-02473-t009:** Risk factors of multiple metastasis on multivariate logistic regression test.

Factors		Multiple Metastasis			
		Initial Recurrence		Final Recurrence	
		HR (95% CI)	*p*	HR (95% CI)	*p*
Pathology	AD	-		2.06 (1.14–3.69)	0.016
	Others			1	
Lymphatic permeation	Present	-		1.75 (1.00–3.03)	0.048
	absent			1	
Pathological stage	II–III	2.57 (1.23–5.36)	0.012	-	
	I	1			

AD, adenocarcinoma; HR, Hazard ratio; 95% CI, 95% confidence interval.

**Table 10 jcm-09-02473-t010:** Multivariate analysis of time to recurrence and survival after recurrence according to the clinicopathological factors.

Multivariate Analysis								
Factors		*n*	Time to Recurrence			Survival after Recurrence		
			Median Months (Range)	HR (95% CI)	*p*	Median Months (Range)	HR (95% CI)	*p*
Age	≥70	110	12 (1–95)	-		13 (0–83)	1.99 (1.39–2.85)	0.001
	<70	113	12 (1–119)			25 (0–182)	1	
Pathology	AD	154	17 (1–119)	-		23.5 (0–180)	0.51 (0.36–0.74)	0.001
	others	69	10 (1–95)			10 (1–182)	1	
Vascular invasion	present	175	11 (1–107)	1.56 (1.12–2.17)	0.008	17 (0–182)	1.61 (1.00–2.36)	0.048
	absent	48	24 (1–119)	1		21 (1–121)	1	
Invasive size	>3 cm	138	11 (1–119)	1.37 (1.03–1.82)	0.028	17 (0–182)	-	
	≤3 cm	85	19 (1–107)	1		21 (0–121)		
Pathological stage	II–III	160	11 (1–80)	1.43 (1.04–1.96)	0.027	17 (0–182)	-	
	I	63	23 (3–119)	1		21 (1–115)		

AD, adenocarcinoma; HR, Hazard Ratio; 95% CI, 95% confidence interval.

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
