# Peer review of "Clinicopathological Factors Related to Recurrence Patterns of Resected Non-Small Cell Lung Cancer"

_jcm, 2020, doi:10.3390/jcm9082473_

Round 1

Reviewer 1 Report

This retrospective work investigates the relationship between a set of clinicopathological factors and recurrence patterns after complete resection of NSCLC as well as the prognostic impact of these factors on time to recurrence and survival. While many similar studies on NSCLC recurrence are already present in the literature, this work adds to our current knowledge by systematically analysing the actual organ-specific patterns of recurrence in the most commonly affected organs (mediastinal lymph nodes, lungs, brain, bone, liver and adrenal glands). Risk factors for organ-specific metastasis and early recurrence are identified, which might allow to better tailor the follow-up after resection in terms of organs to monitor more closely and frequency of follow-up, respectively.

While the novelty is only moderate, a strength of this work is that a thorough analysis of organ-specific recurrence patterns has been done at initial and final recurrence both for all cases as well as the subset of cases with recurrences. As pointed out by the authors, the latter is important as these cases are confirmed to have the potential to metastasize. 

The statistical analysis of the data is scientifically sound and the conclusions drawn are sufficiently supported by the results. Moreover, the limitations of the study are clearly stated.

The readability of the paper suffers from the long tables that extend over several pages. Tables 3 (B) and 4 (B) could probably be shortened. Regarding the other tables, the authors might want to visually highlight the statistically significant results or opt to selectively show the statistically significant results, while moving summary tables with the results for all clinicopathological factors to the supplementary information.

Author Response

Our article: jcm-865119 entitled "Clinicopathological factors related to recurrence patterns of resected non-small cell lung cancer"

We wish to express our deep appreciation to the reviewers for their insightful comments on our paper. We feel the comments have helped us significantly improve the paper. We have revised our previous manuscript and tables in accordance with the reviewer’s comments and suggestions. We believe our revised article will be of special interest to the readers of Journal of Clinical Medicine.

Your comment: The readability of the paper suffers from the long tables that extend over several pages. Tables 3 (B) and 4 (B) could probably be shortened. Regarding the other tables, the authors might want to visually highlight the statistically significant results or opt to selectively show the statistically significant results, while moving summary tables with the results for all clinicopathological factors to the supplementary information.

Reply: Thank you for your comment and suggestion. We all agree on your idea. Although table 3 has been shorted, removing "n.s." data from multivariate analyses data. Table 2 and 4 have been shorten as moving chi-squared test-based data in table 2 to Supplementary table 1 (table S1) and univariate analyses data in table 4 to Supplementary table 2 (table S2). 

Reviewer 2 Report

Clinico-pathological factors related to recurrence patterns of resected non-small cell lung cancer.

by Shimizu et al.

Overall impression.

The study has found results of general interest. For a non-native English-speaking reader, the language seems very good. I have not found anything requiring major revisions.

Specifics.

The motivation for the study given in lines 39-41 is not really fulfilled as the authors do not conclude anything from the results regarding whether shorter period systemic evaluation should be performed during follow-up for a certain subgroup of patients. This should be included in the final conclusion.

The authors do not specify with which methods the diagnostic work-up of these patients had been performed. Were all patients examined with FDG-PET/CT? Did they all undergo bronchoscopic and mediastinal endoscopy? Were they all discussed at a multi-disciplinary team conference before therapy? This should be specified as it has major impact on the interpretation of the findings.

I would facilitate the reading of the table 1 if the authors would also write the fractions or percentage of for instance smoking or recurrence among patients with pleural invasion as the reader then would not have to do the calculations.

Line 78-79: I assume the patients were examined with contrast-enhanced CT (CE-CT), but it should be specified.

Line 82-84: I assume that the PET/CT was performed if any metastasis was discovered and not just in case of a brain metastasis. Again, it should be made clear to avoid misinterpretations.

For table 2 it would as for table 1 facilitate the reading of the table it the authors would also write the percentage of recurrence for instance for males and females etc.

For the HR-results of the multivariate logistic regression it is not necessary to report the results with 3 decimals. One or at most 2 decimals would be sufficient.

Unfortunately figure 1 was not included in the copy of the manuscript that I had for the review.

I noticed two small typing errors in line 202 and 216.

Author Response

Our article: jcm-865119 entitled " Clinicopathological factors related to recurrence patterns of resected non-small cell lung cancer "

We wish to express our deep appreciation to the reviewers for their insightful comments on our paper. We feel the comments have helped us significantly improve the paper. We have revised our previous manuscript and tables in accordance with the reviewer’s comments and suggestions. We believe our revised article will be of special interest to the readers of Journal of Clinical Medicine.

  1. The motivation for the study given in lines 39-41 is not really fulfilled as the authors do not conclude anything from the results regarding whether shorter period systemic evaluation should be performed during follow-up for a certain subgroup of patients. This should be included in the final conclusion.

Reply 1: Thank you for this comment. As you pointed out, we did not mention our claim regarding follow-up planning in conclusion although we had some comments on it in discussion part. We have corrected the final conclusion as below.

Changes in text: In conclusion, the results of the current study demonstrate that some clinicopathological factors were associated with organ-specific metastasis, number of metastasis sites, and post-recurrence survival. This information is important for determining which patients need follow-up with shorter period systemic evaluation and further studies regarding the molecular mechanisms that lead to organ-specific metastasis in NSCLC. (Page 13, Lines 259-263)

  1. The authors do not specify with which methods the diagnostic work-up of these patients had been performed. Were all patients examined with FDG-PET/CT? Did they all undergo bronchoscopic and mediastinal endoscopy? Were they all discussed at a multi-disciplinary team conference before therapy? This should be specified as it has major impact on the interpretation of the findings.

Reply 2: Thank you for pointing out. As your comments, we always discuss all treatments for lung cancer at a multi-disciplinary team conference, taking results of HRCT and PET-CT into account. We have added some sentences in 2.1. Patient characteristics.

Changes in text: Between January 2004 and December 2016, a total of 688 patients with NSCLC underwent complete resection and systematic lymph node dissection at our institute. For diagnostic work-up, all patients underwent chest contrast-enhanced HRCT and positron emission tomography (PET) CT. When nodal metastasis was suspected, additional examination was performed such as endobronchial ultrasound-guided transbronchial needle aspiration or biopsy using mediastinoscopy. All therapies including surgery, adjuvant chemotherapy, and post-recurrent therapies were discussed and decided at a multi-disciplinary team conference. Among them, 223 patients had postoperative recurrence of NSCLC. Patients were excluded if they underwent sublobar resection and/or neoadjuvant chemotherapy and had NSCLC with positive intraoperative pleural lavage cytology. The detailed data of these enrolled patients are shown in Table 1. (Page 2, Lines 52-61)

  1. I would facilitate the reading of the table 1 if the authors would also write the fractions or percentage of for instance smoking or recurrence among patients with pleural invasion as the reader then would not have to do the calculations.

Reply 3: Thank you for your comment. We have added the percentage number in every row of table 1.

  1. Line 78-79: I assume the patients were examined with contrast-enhanced CT (CE-CT), but it should be specified.
  2. Line 82-84: I assume that the PET/CT was performed if any metastasis was discovered and not just in case of a brain metastasis. Again, it should be made clear to avoid misinterpretations.

Reply 4-5: Thank you for this comment. We have clarified these points as below.

Changes in text: All patients were followed-up at our outpatient department quarterly in the first 2 years after resection and semi-annually thereafter. Contrast-enhanced CT scans of the chest and upper abdomen were routinely obtained during every scheduled outpatient visit for follow-up. CT or magnetic resonance imaging (MRI) of the brain was also routinely performed semi-annually. Such radiological examinations were also performed when neurological symptoms occurred or when clinical suspicions were raised. Once any metastasis was discovered, a routine examination such as PET was performed to determine the presence of other metastatic sites, if any. (Page 3, Lines 82-88)

  1. For table 2 it would as for table 1 facilitate the reading of the table it the authors would also write the percentage of recurrence for instance for males and females etc.

Reply 6: Thank you so much for your advice. As in table 1, we have added the percentage number in every row. Please note that former table 2 has been divided into 2 parts (new table 2 and supplementary table 1) due to first reviewer’s suggestion.

  1. For the HR-results of the multivariate logistic regression it is not necessary to report the results with 3 decimals. One or at most 2 decimals would be sufficient.

Reply 7: Thank you for your comment. We have corrected HR numbers with 2 decimals in tables as well as in text.

  1. Unfortunately figure 1 was not included in the copy of the manuscript that I had for the review.

Reply 8: We do not know why figure 1 was not seen in our manuscript. The revised manuscript includes figure 1 in page 11.

  1. I noticed two small typing errors in line 202 and 216.

Reply 9: Thank you for pointe out typos. We have replaced the term “orangs” with “organs” in former line 202. Regarding line 216, we were not able to find apparent typos. But, to precisely express our claim, we have rewritten the last sentence in first paragraph of discussion as below.

Changes in text: Although there were no apparent differences in the risk factors for the initial and final recurrence sites, to the best of our knowledge, the current study is the first to evaluate the metastatic lesions until the end of the follow-up period. (Page 11, Lines 203-205)